# Uncovering Time-Invariant Latent Representation for Brain Disorder Diagnosis via Self-Supervised Learning

## Abstract

Recently, large-scale deep-learning models and datasets have shifted the development of medical image analysis with robust and generalizable representations. In this context, self-supervised learning has emerged as a valuable tool, offering the advantage of advancing deep learning without the need for costly annotations while facilitating downstream tasks with limited sample sizes. However, this feature has been few investigated in brain network analysis, and most existing self-supervised learning approaches yield only comparable performances with those achieved without self-supervised learning. In this study, we introduce an efficient self-supervised representation learning approach known as Bootstrap Time-Invariant Latent (BTIL), aiming at capturing time-invariant representations of brain networks derived from resting-state fMRIs for the diagnosis of brain disorders. We randomly dropped some timepoints in the functional signals and subsequently derived two augmented pseudo-functional connectivity (pFC) as positive pairs. Our BTIL consists of an online network and a target network, where each network encodes one augmented pFC. The time-invariant representations are obtained by bringing the latent embeddings of the two networks closer. Additionally, we employ Mask-ROI Modeling (MRM) with both classification and reconstruction heads for relating intra-network dependencies and enhancing regional specificity. Linear evaluations on three downstream classifications demonstrate the superiority of BTIL for brain disorder diagnosis with more than 2% improvements compared with the state-of-the-art works.

## 1 Introduction

Functional neuroimaging, e.g. functional Magnetic Resonance Imaging (fMRI), using the blood-oxygenlevel-dependent (BOLD) effect has received considerable attention in the past decades and has become a powerful tool in neuroscience Sato et al. (2006); Bullmore & Sporns (2009); Racine et al. (2005). This paradigm offers a unique opportunity to map the neural substrates of cognition in-vivo. One crucial outcome is the functional brain networks. These networks are established through pairwise correlations between the BOLD signal time series extracted from various regions of interest (ROIs). Recently, they have recently represented an indispensable foundation for neuroscience studies by characterizing the complex relationship between brain dysfunctions and phenotypes Bargmann & Marder (2013); Buckner et al. (2013). Gaining a deeper understanding of the communication between brain regions is both from the perspective of understanding how our brain facilitates higher-order cognition and also to provide insight into how brain disorders arise.

In recent years, deep learning approaches have significantly influenced the field of brain functional network analysis. These approaches encompass various techniques, ranging from convolutional neural networks (CNN) Kawahara et al. (2017); Huang et al. (2017; 2020), graph neural networks (GNN) Zhao et al. (2022); Li et al. (2021); Yang et al. (2023a), to Transformer networks Kan et al. (2022b); Zhu et al. (2022). Many of these studies involve constructing brain networks by calculating the Pearson correlation between regional activation time series over the full scan duration, which is referred to as static functional connectivity (sFC). Another approach is to build brain networks using shorter sliding windows, known as dynamic functional connectivity (dFC). Sequential modeling with dFC on shorter sliding windows provides additional dynamic activation information compared

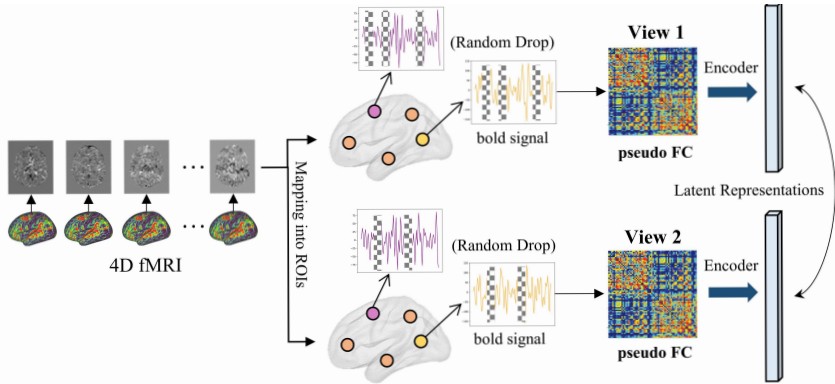

Figure 1: Illustration of our study behind time-invariant representation learning.

to models using sFC. However, when applied to resting-state fMRIs, these studies achieve only comparable or sometimes even lower performance than those using sFC, particularly in the context of brain disease diagnosis Kim et al. (2021); Kan et al. (2022a); Azevedo et al. (2022); Bedel et al. (2023). One main reason for this is that these models incorporate more redundant features while working with limited data samples. Previous studies have suggested that dFC can preserve similar topological properties to sFC Allen et al. (2014); Sakoğlu et al. (2010); Hindriks et al. (2016) with a proper sliding setting. Based on these observations, we hypothesize that demographic factors such as age, sex, and diagnosis are not highly sensitive to the duration of time in resting-state fMRIs. Consequently, the main challenge in diagnosing brain disorders lies in how to extract time-invariant disease-specific representations underlying neuroimages.

To achieve this, in this study, we introduce a self-supervised learning scheme to uncover time-invariant latent representations for brain disorder diagnosis. As is shown in Figure 1, we randomly drop some timepoints of the whole BOLD signals and derive new views of brain networks as positive pairs, termed pseudo functional connectivity (pFC). These positive pairs can be viewed as a data augmentation of sFC. We extract the latent representations of these pFCs by a Transformer encoder and a projection, and supervise the network to bring the representations of different pFCs closer. In this manner, the latent embeddings tend to be time-invariant and disease-specific. The framework is represented as Bootstrap Time-Invariant Latent (BTIL), which is constructed in a self-supervised learning manner, inspired by previous works Grill et al. (2020); Richemond et al. (2020). Compared with most contrastive learning methods Chen et al. (2020a;b); He et al. (2020), BTIL is trained without using negative pairs, instead bootstrapping the representations by learning from an online network and a target network. Starting from a pFC, BTIL trains its online network to predict the target network's representation of another pFC. Moreover, we implement the Masked ROI Modeling (MRM) with both ROI classification and feature reconstruction heads into the online network. This inclusion helps relate intra-network dependencies and enhances local specificity for classification tasks. The regional distinguishing capability is particularly valuable when dealing with disorders exhibiting significant intra-class variations, especially in cases of psychological disorders like autism spectrum disorder (ASD) and attention deficit hyperactivity disorder (ADHD).

We access the representations on three real-world datasets including ABIDE, ADHD-200, and REST-MDD. These evaluations involve the diagnosis of ASD, ADHD, and Major Depression Disorder (MDD) from healthy controls (HC). It's worth noting that diagnosing these three psychiatric disorders is particularly challenging due to the substantial intra-class variations among patients. In our downstream linear evaluations, which entail training a linear classifier on the frozen representations, BTIL consistently outperforms the current state of the art by more than 2% across all three datasets. In summary, our study is structured around addressing three research questions (RQ):

- RQ1: Are time-invariant representations reasonable for disease diagnosis?

- RQ2: How powerful is BTIL in diagnosing multiple types of brain disorders?

- RQ3: How do different parameter settings, such as model size and mask ratio, impact the framework?

To address RQ1, we examined the time-invariant characteristics of BOLD signals and assessed various ways for augmenting brain networks in Section 4.2. In Section 4.3, we present our findings on the superior classification performance across the three public datasets, addressing RQ2. For RQ3, we conduct a sensitivity analysis and ablation studies, which are detailed in Section 4.4. Finally, in Section 5, we discuss the potential limitations of our framework and outline directions for future research and model development.

## 2 RELATED WORKS

**Brain Connectome Studies.** Significant advancements have been made over the past decade in the application of neuroimaging techniques to uncover alterations in brain networks associated with various brain disorders. Convolutional neural networks (CNN) are firstly proposed to facilitate end-to-end disease identification with promising performances and have been widely applied for analyzing connectome patterns such as BrainNetCNN Kawahara et al. (2017) and Deep Convolutional Auto-Encoder Huang et al. (2017). In addition to CNNs, graph neural networks (GNNs) have gained prominence. GNNs have the capacity to capture information about neighboring structures within the brain. BrainGNN, for instance, introduced ROI-aware graph convolutional layers and ROI-selection pooling layers to predict neurological biomarkers at both the group and individual levels Li et al. (2021). Another approach, proposed by Ktena et al. (2018), involved learning a graph similarity metric using a siamese graph convolutional neural network. Xing et al. (2019) introduced a dynamic spectral graph convolution network, which constructed connectivity patterns from time-varying correlations in fMRI signals. In a similar vein, Zhao et al. (2022) trained a dynamic graph network by learning from sparse connections among brain regions calculated dynamically from graph features. More recently, the Transformer architecture has garnered considerable attention due to its exceptional performance in graph representation learning. However, most existing Transformer-based networks Ying et al. (2021); Kreuzer et al. (2021); Dwivedi & Bresson (2020) have achieved only limited success in brain network analysis. To address this limitation, BrainNetTransformer Kan et al. (2022b) was introduced to harness the potential of Transformer-based models for enhanced brain network analysis.

**Self-supervised Learning.** Self-supervised learning paradigms have delivered promising results in computer vision He et al. (2020); Chen et al. (2020b;b); Fan et al. (2021); He et al. (2022) and natural language processing Devlin et al. (2018); Radford et al. (2018); Wu et al. (2021); Conneau & Lample (2019). These paradigms have introduced pre-trained foundation models that leverage self-supervised learning on extensive unannotated data. This approach produces standardized and generalized representations, offering substantial benefits across domains with limited task-specific data availability. However, this paradigm has been few studied for brain networks. BrainNPT Hu et al. (2023), for example, constructs disturbance inputs by replacing regional features to enhance the models' understanding of the underlying input patterns. BrainGSLs Wen et al. (2023), on the other hand, proposes an ensemble masked graph self-supervised framework based on masking and prediction. Nevertheless, these two studies have only achieved modest improvements when compared to approaches without pre-training (i.e., approximately 71.5% accuracy on the ABIDE dataset). It's important to note that these pre-training strategies, borrowed from BERT-like models, still rely on a substantial amount of training data to establish data dependencies, which may not be suitable for brain network studies. In summary, there remains a notable gap in the development of self-supervised learning strategies tailored to uncover the intrinsic characteristics of brain networks.

## 3 TIME-INVARIANT BRAIN CONNECTOME SELF-SUPERVISED LEARNING

### 3.1 PROBLEM DEFINITION

Our objective is to develop a mapping function $f : \boldsymbol{X} \rightarrow y$, where $\boldsymbol{X} \in \mathbb{R}^{V \times V}$ represents a brain network with $V$ Regions of Interest (ROIs), and $y$ denotes the predicted diagnosis phenotype for each subject. Our approach consists of two key phases: self-supervised representation learning and downstream classification. The self-supervised learning framework is depicted in Figure 2 (a), which comprises an online network, a target network, and a prediction function. Both the online and target networks share the same architecture, featuring a Transformer network and a readout function. In the online network, input brain networks undergo encoding via an $L$-layer self-attention

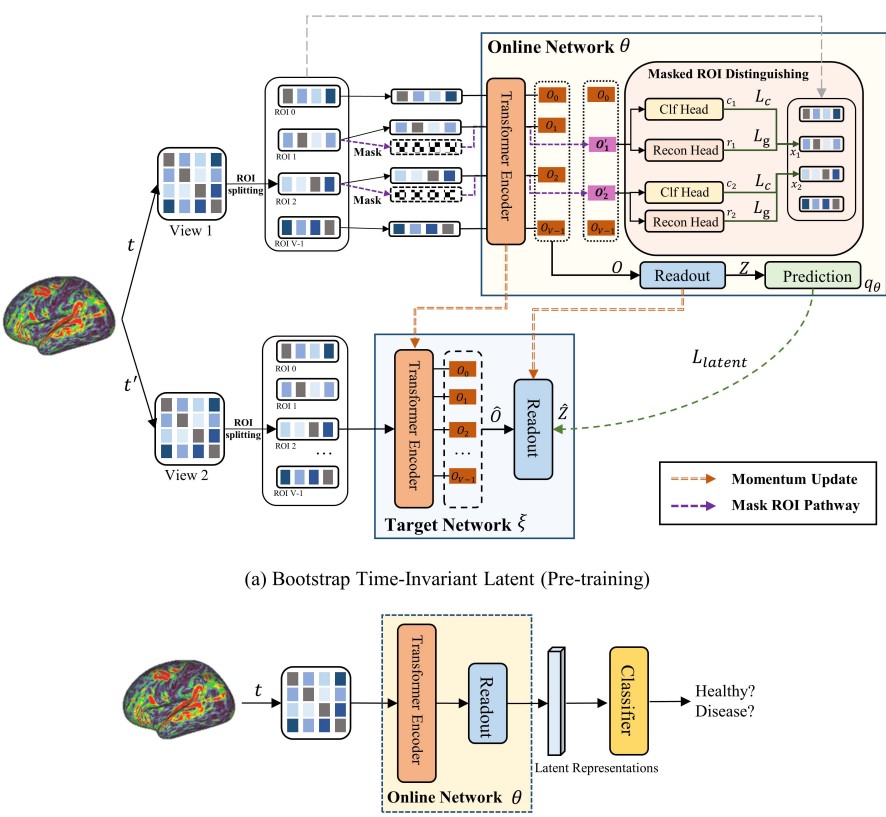

(a) Bootstrap Time-Invariant Latent (Pre-training)

(b) Transfer-learning for downstream disease classification

Figure 2: The overall framework of BTIL comprises two phases. In the initial phase, the model undergoes self-supervised learning. In the subsequent phase, the acquired representations are input into a classifier for disease diagnosis prediction. The components are color-coded for clarity: the yellow and blue segments represent the online network and the target network, respectively, while the pink section symbolizes the Masked ROI Modeling module.

Multi-head Self-Attention (MHSA) Transformer network, leading to nonlinear mappings denoted as $\boldsymbol{X} \rightarrow \boldsymbol{O} \in \mathbb{R}^{V \times V}$. Subsequently, the readout function transforms the encoded features $\boldsymbol{O}$ into subjective embeddings $\boldsymbol{Z} \in \mathbb{R}^{D \times V}$. Similarly, the target network generates subjective embeddings $\hat{\boldsymbol{Z}}$ using the same process. Finally, a prediction MLP is employed to learn the mapping from the online network outputs $\boldsymbol{Z}$ to predict the target network outputs $\hat{\boldsymbol{Z}}$. In the downstream classification phase, a Support Vector Machine (SVM) classifier is employed for prediction. This classifier takes input features inferred from the learned representations $\boldsymbol{Z}$ derived from the online network.

## 3.2 TIME-INVARIANT BRAIN NETWORK AUGMENTATION

In this study, we propose to investigate brain network augmentation methods involving the random removal of certain timepoints within timeseries data. Given a brain network $\boldsymbol{X}$ derived from a timeseries matrix $\boldsymbol{S} \in \mathbb{R}^{V \times T}$ with $T$ time steps, we randomly generate a mask vector $\boldsymbol{m} \in \mathbb{R}^{M}$, where $M \leq V$, and utilize it to exclude these masked timepoints. We then apply Pearson correlation to the modified timeseries matrix $\hat{\boldsymbol{S}} \in \mathbb{R}^{V \times (T-M)}$ to generate pFC $\hat{\boldsymbol{X}}$. It's crucial to acknowledge that fMRI data is frequently acquired using diverse protocols, leading to variations in scanning durations. Consequently, the length of timeseries data may differ between individual samples. To accommodate this variability, we explore various mask lengths, considering different numbers of time steps and different percentages of time steps. The impact of these masking configurations will be addressed in Section 4.2.

### 3.3 SELF-SUPERVISED LEARNING FRAMEWORK

**Latent representation learning.** BTIL is designed to learn a time-invariant latent representation $\boldsymbol{Z}$ containing specific embeddings for downstream diagnosis tasks by connecting two views of augmented brain networks. Following previous works Grill et al. (2020); Chen et al. (2020b), we employ two neural networks: the online network and the target network. Both networks share the same architecture, featuring a Transformer encoder and a readout function. The target network provides regression targets for training the online network. Its parameters $\xi$ are updated using a momentum approach based on the online parameters $\theta$, as $\xi \leftarrow \tau\xi + (1-\tau)\theta$, where $\tau$ is a target decay rate $\tau \in [0,1]$. To prevent collapsed solutions Grill et al. (2020), a predictor $q_\theta$ is applied to the online network to predict the target outputs. The optimization is performed using the mean squared error between the normalized predictions and the target projections:

$$L_{latent} = 2 - 2\frac{<q_\theta(\boldsymbol{Z}), \hat{\boldsymbol{Z}}>}{||q_\theta(\boldsymbol{Z})||_2 \cdot ||\hat{\boldsymbol{Z}}||_2} \tag{1}$$

**Transformer encoder.** Transformer-based models have led a tremendous success in various downstream tasks across fields including natural language processing, computer vision, and also graph learning. However, the brain network data potentially falls in neither of these classes. The brain networks are symmetric semi-positive defined matrices and densely distributed. Previous studies tackle the brain networks as graph data, however, there are still no explicit relationships between ROIs Kan et al. (2022b); Yang et al. (2023b); Li et al. (2021). In this study, we instead tackle the brain network connection profile as a sequence, where each ROI is represented as a sequential step with $V$ features. The input brain network $\boldsymbol{X}$ is viewed as a sequence $\{\boldsymbol{x}_0, \boldsymbol{x}_1, ..., \boldsymbol{x}_{V-1}\}$, where the $i$-th element is obtained by $\boldsymbol{x}_i = \boldsymbol{X}_{i,:} \in \mathbb{R}^V$. In this context, Multi-Head Self-Attention is implemented to relate inter-ROI dependencies and generate more expressive brain features $\boldsymbol{H}^L = \text{MHSA}(\boldsymbol{X}) \in \mathbb{R}^{V \times V}$. For each layer $l$, we first calculate the query $\boldsymbol{Q}^{l,c}$, key $\boldsymbol{K}^{l,c}$, and value $\boldsymbol{V}^{l,c}$ for the $c$-th head through linear projection as $\boldsymbol{Q}^{l,c} = \boldsymbol{H}^{l-1}\boldsymbol{W}_q^{l,c}$, $\boldsymbol{K}^{l,c} = \boldsymbol{H}^{l-1}\boldsymbol{W}_k^{l,c}$, $\boldsymbol{V}^{l,c} = \boldsymbol{H}^{l-1}\boldsymbol{W}_v^{l,c}$, where $\boldsymbol{H}^{l-1}$ is the output of the $l$-th layer, $\boldsymbol{H}^0 = \boldsymbol{X}$, and $W_q^{l,c}, W_k^{l,c}, W_v^{l,c}$ are learnable parameters. $c$ is in the range of $\{1, 2, ..., C\}$, and $C$ denotes the number of self-attention heads. The output for each head is computed as a scaled dot-product as $\boldsymbol{H}^{l,c} = \text{Softmax}(\frac{\boldsymbol{Q}^{l,c}(\boldsymbol{K}^{l,c})^T}{\sqrt{d}})\boldsymbol{V}^{l,c}$, where $d$ is the first dimension of $\boldsymbol{W}^{l,c}$. Finally, the output $\boldsymbol{H}^l$ is obtained by $\boldsymbol{H}^l = (||_{c=1}^C \boldsymbol{H}^{l,c})\boldsymbol{W}_O^l$, where $||$ is the concatenation operator, and $\boldsymbol{W}_O^l$ are learnable model parameters. We implement the Feed Foward Network and layer normalization for mapping the $\boldsymbol{H}^l$ into encoder outputs $\boldsymbol{O}$.

**Readout function.** After obtaining the non-linear features from the Transformer encoders, we are left with high-dimensional data, which can pose challenges for downstream classification tasks, especially given the limited fMRI data samples available. In this study, we employ a readout function to transform the output features $\boldsymbol{O}$ into subject-specific embeddings $\boldsymbol{Z}$. To achieve this, we aggregate the output features for each ROI into a set of $D$ features. These features are then concatenated to form the final feature representation $\boldsymbol{O} \in \mathbb{R}^{D \times V}$. In this study, we set $D = 8$ and obtain $8 \times 100 = 800$ representations for each brain network.

**Masked ROI distinguishing.** In this study, we introduce a joint discriminative and generative objective to regulate the online network's pretraining process and establish intra-network ROI dependencies. As illustrated in Figure 2, each input brain network is treated as a sequence, divided into $V$ ROIs, and randomly assigned a set $\mathbb{P}$ of $P$ masked ROI position indices. For each patch that needs to be masked, we replace its patch embedding with a learnable mask embedding. Positional embeddings are added to the patch embeddings, and the resulting data is fed into the Transformer encoder. For each masked patch $\boldsymbol{x}_i$, we obtain a corresponding output $\boldsymbol{o}_i'$ from the Transformer encoder. Subsequently, we pass $\boldsymbol{o}_i'$ through both a classification head and a reconstruction head to obtain outputs $\boldsymbol{c}_i$ and $\boldsymbol{r}_i$, respectively. Both the classification and reconstruction heads consist of two-layer MLPs designed to map $\boldsymbol{o}_i'$ to the same dimension as $\boldsymbol{x}_i$. The goal is to make $\boldsymbol{r}_i$ as close as possible to $\boldsymbol{x}_i$ while ensuring the model can correctly match pairs $(\boldsymbol{x}_i, \boldsymbol{c}_i)$. To achieve this, we employ the InfoNCE loss Oord et al. (2018) $L_c$ for the classification objective and the mean square

error loss $L_r$ for the reconstruction objective:

$$L_c = -\frac{1}{N} \sum_{i=1}^{N} \log \left( \frac{\exp(\boldsymbol{c}_i^T \boldsymbol{x}_i)}{\sum_{j=1}^{N} \exp(\boldsymbol{c}_i^T \boldsymbol{x}_j)} \right) \tag{2}$$

$$L_r = \frac{1}{N} \sum_{i=1}^{N} (r_i - x_i)^2 \tag{3}$$

To note that, there remains two types of inputs for the online network training. For latent representation learning, non-masked brain networks are fed into the online network to obtain the output features $\boldsymbol{O}$, while for the MRM module, the online network encodes the masked brain networks to prediction the mask embeddings. Finally, the objective function for self-supervised pre-training is obtained through a weighted summation, incorporating $\lambda_c$ and $\lambda_r$ to balance the training:

$$L = L_{latent} + \lambda_c L_c + \lambda_r L_r \tag{4}$$

## 4 EXPERIMENTS AND RESULTS

### 4.1 EXPERIMENTAL SETTINGS

**Datasets.** We evaluated our approach on three real-world datasets, and their demographic information is provided in Appendix A. Here's a brief overview of the datasets: (a) **ABIDE (Autism Brain Imaging Data Exchange-I)**: The ABIDE-I dataset included 528 individuals diagnosed with ASD and 555 controls, with fMRI data collected from 17 international sites. All images underwent preprocessing using the C-PAC pipeline and were mapped into the Schaefer-100 atlas, consisting of 100 brain regions. (b) **ADHD-200 (ADHD-200 global competition)**: The ADHD-200 dataset comprises 710 healthy control subjects and 550 patients with ADHD. Data from eight international sites were used, and similar to ABIDE, preprocessing and mapping into the Schaefer-100 atlas were performed. (c) **REST-MDD**: The REST-MDD dataset includes 1276 patients with depression and 1104 healthy controls from 25 international sites. Harvard-Oxford atlas was used for mapping fM-RIs into 112 brain regions. In all three datasets, the repetition time (TR) was 2 seconds. Further details regarding preprocessing can be found in Appendix B.

**Implementation details.** To ensure comparability, we employed a consistent approach for dataset splitting across all three datasets: 70% for training, 15% for validation (if necessary), and 15% for testing. The training set served for both self-supervised pre-training and classifier supervision. We utilized the Adam optimizer with an initial learning rate of $3 \times 10^{-5}$ and a weight decay of $5 \times 10^{-5}$. The learning rate underwent a linear increase to $3 \times 10^{-4}$ within 10 warmup epochs. The batch size was set to 256 for all datasets. The BTIL model underwent training for 10,000 epochs, and we saved the models with the lowest training loss for subsequent classification tasks. The decay rate for target network update is set as $0.996$ Our experiments were conducted on a platform equipped with 64 NVIDIA Tesla V100 GPUs, with 4 GPUs allocated for each training run. The training duration varied, taking approximately 8, 8, and 12 hours for the ABIDE, ADHD-200, and REST-MDD datasets, respectively. To explore the effects of different model sizes, we configured five parameter settings, including hidden features, Transformer layers, and the number of attention heads, as detailed in Appendix C. For the loss function, as shown in Eq. 4, we set $\lambda_c$ to 0.1, which had minimal impact on downstream classification tasks. The discussion on the choice of $\lambda_r$ can be found in Section 4.4. For the downstream classification tasks, the encoded latent representations were input into an SVM classifier for prediction.

**Metrics.** We assess the performance of diagnosis classification, differentiating ASD, ADHD, and MDD from HC across the three datasets, using accuracy, sensitivity, and specificity as our key metrics. Both tasks involve binary classification. To account for the variability introduced by data collection from multiple centers, we employ a rigorous stratified sampling strategy that considers collection sites during the training-validation-testing split. This approach aligns with previous studies, ensuring fair comparisons Kan et al. (2022b); Parisot et al. (2018). All reported performance metrics represent averages computed over 10 random test set runs, accompanied by standard deviation values. Our code and pre-trained checkpoints are publicly available online (currently anonymous).

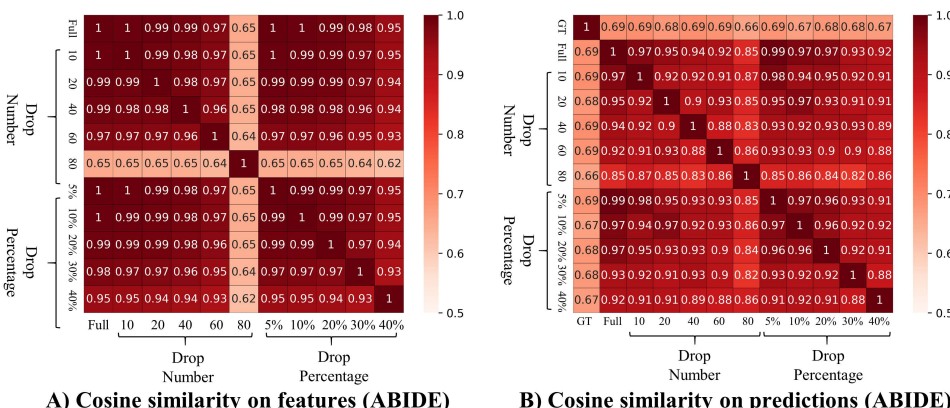

Figure 3: Evaluation on the feature similarity and prediction similarity of time-invariant augmentation. GT: ground truth. Full: raw features derived from the full timesteps.

## 4.2 TIME-INVARIANT FEATURE EVALUATION (RQ1)

We embarked on an assessment of two distinct time-invariant data augmentation techniques, specifically random time-point removal in fixed and unfixed settings. Cosine similarity served as our metric for gauging both feature similarity (pre-hoc) and prediction similarity (post-hoc), as elucidated in Figures A) and B) of Figure 5. Supplementary results for the ADHD-200 and REST-MDD datasets are accessible in Appendix E.1. The pre-hoc analysis primarily centers around evaluating feature similarity within the brain network. Our results reveal that judiciously configured random drop augmentation can yield robust and closely aligned features. Furthermore, whether the number of dropped time-points is fixed ("drop by number") or unfixed ("drop by percentage"), both strategies yield analogous outcomes. Notably, increasing the number of dropped time-points consistently leads to a reduction in similarity. In the post-hoc analysis, we shift our focus to assessing prediction similarity using SVM classification. For this analysis, we flatten the brain network features into a vector comprising $V \times (V - 1)/2$ elements. Remarkably, the post-hoc results align closely with those from the pre-hoc analysis. Additionally, we conducted experiments involving time-invariant data augmentation with BTIL, as elaborated in Appendix E.2. In summary, our findings suggest that configurations involving the random removal of fewer than 40 timesteps or 20% represent preferable choices. These results underscore the advantages of time-invariance in resting-state fMRI, which can be harnessed as a data augmentation strategy and for enhancing disease diagnosis.

## 4.3 BRAIN DISORDER DIAGNOSIS PERFORMANCE (RQ2)

To assess the performance of brain disorder diagnosis, we conducted comparisons involving BTIL against two categories of baseline models: those with self-supervised learning (SSL) and those without SSL. The baseline models without SSL include BrainNetCNN Kawahara et al. (2017), DHGNN Jiang et al. (2019), BrainGNN Li et al. (2021), Semi-GCN Parisot et al. (2018), vanilla-Transformer (vanillaTF), and BrainNetTransformer (BrainNetTF) Kan et al. (2022b). For SSL comparisons, the architecture of BTIL closely resembles that of BYOL Grill et al. (2020) and MOCO He et al. (2020), which are also included in the evaluation. Furthermore, we considered two existing works: BrainNPT Hu et al. (2023) and BrainGSLs Wen et al. (2023). Detailed implementations of these comparable approaches can be found in Appendix D.

Table 1 displays the results, with the best performance shown in bold and the second best underlined. Key observations are as follows: 1) Among the baseline models without pretraining, CNN, GNN, and Transformer models achieve comparable performances across all three datasets. Transformer models, with increased computational complexity, exhibit limited performance gains given the fMRI data in limited samples. However, BrainNetTF, incorporating orthonormal clustering readout, significantly enhances diagnosis performance, aligning with previous studies Kan et al. (2022b). 2) In contrast, most self-supervised learning approaches did not contribute significantly to performance improvement when compared to BrainNetTF. This limitation can be attributed to the

scarcity of available data samples. It's worth noting that, as reported in Hu et al. (2023), Brain-NPT achieves a performance of 71.25% on the ABIDE dataset when pre-trained on a broad dataset, including ABIDE, HCP, and REST-MDD datasets. Such self-supervised learning paradigms still heavily depend on large-scale data, which is not available for fMRI studies. Overall, these existing self-supervised learning approaches are limited in their performance. 3) Lastly, our proposed BTIL achieves consistent improvements across all three datasets compared to state-of-the-art approaches (with improvements of 2.06%, 2.92%, and 3.06% in accuracy for ABIDE, ADHD-200, and REST-MDD datasets, respectively). This underscores the crucial role of time-invariant representations in brain disorder diagnosis and the framework for self-supervised learning.

Table 1: Classification results of different approaches on three datasets (ABIDE, ADHD-200, and REST-MDD) in terms of accuracy (Acc), sensitivity (Sen), and specificity (Spe).

| Methods | | ABIDE-I | | | ADHD-200 | | | REST-MDD | | |
|---|---|---|---|---|---|---|---|---|---|---|
| Model | Type | Acc | Spe | Sen | Acc | Spe | Sen | Acc | Spe | Sen |
| BrainNetCNN | CNN | 68.14 | 67.56 | 69.74 | 61.62 | 63.18 | 61.82 | 62.55 | 65.41 | 59.93 |
| DHGNN | GNN | 64.31 | 63.81 | 64.97 | 59.84 | 53.52 | 61.72 | 59.24 | 61.40 | 56.51 |
| BrainGNN | GNN | 69.60 | 61.47 | **76.46** | 61.02 | 54.60 | 64.08 | 61.40 | **71.37** | 50.86 |
| PopGCN | GNN | 69.76 | 67.61 | 71.72 | 62.20 | 56.69 | **66.40** | 61.20 | 65.06 | 57.23 |
| vanillaTF | TF | 68.98 | 65.01 | 72.48 | 61.62 | 63.18 | 61.82 | 62.49 | 64.60 | 60.65 |
| BrainNetTF | TF | 71.02 | 73.27 | 71.18 | 62.75 | 63.61 | 62.85 | 63.50 | 65.05 | 61.56 |
| MOCO-V3 | SSL | 70.32 | 70.74 | 71.91 | 63.30 | 64.29 | 63.14 | 63.34 | 62.64 | 64.83 |
| BYOL | SSL | 71.04 | 70.60 | 72.11 | 63.35 | 61.97 | 63.88 | 63.80 | 64.43 | 63.61 |
| BrainNPT | SSL | 68.92 | 67.29 | 70.00 | 62.49 | 59.60 | 63.31 | 57.84 | 58.84 | 56.01 |
| BrainGSLs | SSL | 71.30* | 70.20* | 69.90* | 62.32 | 63.48 | 66.25 | 59.87 | 59.11 | 62.22 |
| BTIL | SSL | **73.36** | **74.02** | 73.12 | **66.27** | **68.61** | 65.70 | **66.56** | 66.59 | **66.58** |

## 4.4 SENSITIVE ANALYSIS AND ABLATION STUDIES (RQ3)

**Ablation studies.** We conducted evaluations on the various components of Eq. 4. These components encompass latent representation learning ($L_{latent}$), classification heads ($L_c$), and reconstruction heads ($L_r$) for ROI distinguishing. We listed the corresponding results in Table 2. The results consistently indicate that the inclusion of the $L_r$ term leads to performance improvements across all three datasets. Although the ROI classification term has a relatively minor impact on performance, combining it with $L_{latent} + L_r$ further enhances model performance. This improvement is attributed to the model's enhanced capability to capture dependencies among brain regions and intra-brain network representation learning facilitated by the masked ROI distinguishing module.

Table 2: Ablation studies on the elements of BTIL.

| | $L_{latent}$ | $L_{latent} + L_c$ | $L_{latent} + L_r$ | BTIL |
|---|---|---|---|---|
| ABIDE | 71.02 | 72.04 | 72.99 | 73.36 |
| ADHD | 63.35 | 63.29 | 64.05 | 66.27 |
| REST-MDD | 62.80 | 62.46 | 66.00 | 66.56 |

Table 3: Effect on $\lambda_r$ values

| | 1 | 5 | 10 | 20 |
|---|---|---|---|---|
| ABIDE | 71.92 | 72.52 | 73.36 | 73.06 |
| ADHD | 63.78 | 65.03 | 66.27 | 64.54 |
| REST-MDD | 62.55 | 62.30 | 66.56 | 62.94 |

**Mask ratio analysis.** In this study, the choice of mask ratio proves to be a pivotal factor within the Masked ROI Modeling module. As illustrated in Figure 4 A), we observe that performance initially improves with an increase in the mask ratio and then starts to decline. Across all three datasets, the optimal mask number is 10 (equivalent to 10% with 100 ROIs) for classification heads and ranges between 5 and 20 (5% to 20%) for reconstruction heads. Performance deteriorates when the mask ratio exceeds 20 across all three datasets. It's worth noting that, in contrast to other frameworks like Masked AutoEncoder He et al. (2022), our BTIL exhibits limitations at high mask ratios.

**Model Size analysis.** Figure 4 B) depicts the accuracy performance of models of different sizes, and the corresponding hyperparameter settings can be found in Appendix C. It is worth noting that the top-performing models are the base model (REST-MDD dataset) and the large model (ABIDE and ADHD-200 datasets). However, as these models incorporate more hyperparameters, their performance starts to plateau due to the constraints posed by the limited availability of data samples.

**The balance of mask & prediction and latent learning.** Throughout our experiments, we noticed a substantial impact on downstream classification performance stemming from the choice of $\lambda_r$. This effect is illustrated in Table 3. When $\lambda_r$ is set to a small value, like $\lambda_r = 1$, the contribution

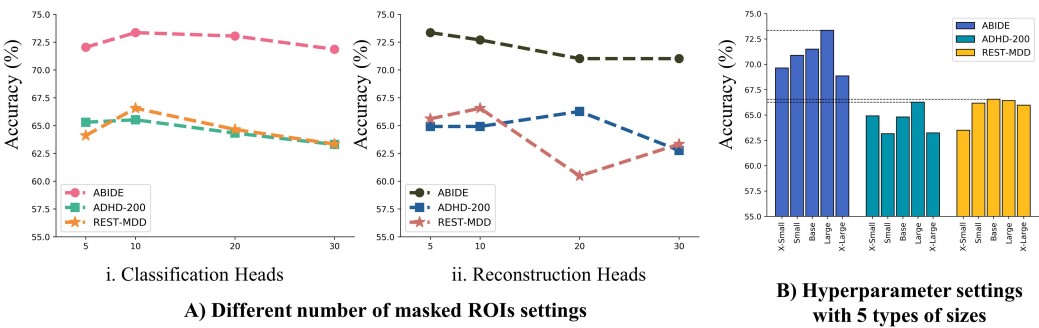

Figure 4: Sensitive analysis on the mask ratio on classification heads (i), reconstruction heads (ii) in A), and hyperparameter settings in B).

of the reconstruction loss to the gradient update becomes minimal, making it challenging to achieve convergence. In this regard, we opted for higher values for the reconstruction loss. The results in the table demonstrate that setting $\lambda_r = 10$ consistently yields top performance across all three datasets.

## 5 DISCUSSION AND CONCLUSION

In this study, we introduce BTIL, a self-supervised approach designed to uncover time-invariant latent representations for diagnosing brain diseases. Our method comprises two essential components: time-invariant data augmentation and a latent representation learning framework featuring a masked ROI distinguishing module, both of which play crucial roles in downstream classifications. This paradigm has led to several significant contributions. Firstly, it has markedly enhanced disease diagnosis performance. To our knowledge, our study represents the first to achieve an accuracy exceeding 73% on the ABIDE dataset, surpassing previous limitations. Secondly, the self-supervised learning paradigm has offered insights into the interpretation of biomarkers with broad data. In Appendix F, we present an evaluation of attention weights, demonstrating that the key regions returned by MHSA align with findings from prior studies.

Nonetheless, neuroscience studies still face substantial challenges, including the collection of large-scale, hard-to-obtain datasets and the high variability in scanning protocols. In this study, we trained our BTIL model on individual database. However, when we attempted to consolidate diverse data for pre-training, the downstream classification performance showed a decrease to some extent, as illustrated in Appendix E.4. This decline can be attributed to both the limited number of data samples and the significant variability in data domains among different data centers. Our future work aims to construct a large-scale, adaptable foundational model for brain disease diagnosis.

In summary, our study introduces BTIL, a self-supervised learning approach tailored for resting-state fMRI data, which surpasses the performance limits of conventional methods by achieving more than a 2% improvement. Our experimental results, spanning three datasets, validate the effectiveness of our framework. This research not only provides valuable data augmentation techniques but also advances the evaluation of training frameworks, offering fresh perspectives on large-scale self-supervised learning for brain functional network analysis.

## 6 REPRODUCIBILITY STATEMENT

Our code is available online publicly (anonymous for now), and is accessable in the supplementary material. We provide the following: (1) Code used to reproduce the results in our paper. (2) README explaining how to install packages, preprocess data, and run experiments. (3) we provide scripts to automate the process of running experiments across the various models, ablation studies, etc. The checkpoint pre-trained on the ABIDE dataset is attached. Other checkpoints would be released online. We hope this will allow others to use our code for future projects.

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

## A   DEMOGRAPHICAL INFORMATION

Table 4: Demographical statistics on the three datasets.

| Dataset | Group | Gender (M/F) | Age (Mean±Std) | Min. Timesteps | Max. Timesteps |
|---------|-------|--------------|----------------|----------------|----------------|
| ABIDE | NC | 491/95 | 16.78±7.71 | | |
| | ASD | 466/62 | 16.95±8.44 | 82 | 320 |
| ADHD | NC | 358/352 | 11.96±3.48 | | |
| | ADHD | 398/124 | 11.26±2.94 | 59 | 261 |
| REST-MDD | NC | 462/641 | 36.15±15.67 | | |
| | MDD | 463/813 | 36.23±14.62 | 90 | 240 |

## B   FMRI DATA PREPROCESSING

All the fMRI images were pre-processed by reference to the Configurable Pipeline for the Analysis of Connectomes (C-PAC) pipeline Craddock et al. (2013), including skull striping, slice timing correction, motion correction, global mean intensity normalization, nuisance signal regression with 24 motion parameters, and band-pass filtering (0.01-0.08Hz). The functional images were finally registered into standard anatomical space (MNI152). The mean time series for a set of regions were computed and normalized into zero mean and unit variance. Pearson Coefficient Correlation was applied to measure functional connectivity.

## C   HYPERPARAMETER SETTINGS

Table 5: Hyperparameter settings for five types of model size.

| | | n_layer | n_heads | FFN dim | MLP dim | #Params |
|---|---|---------|---------|---------|---------|---------|
| | X-Small | 4 | 4 | 1024 | 512 | 7.65M |
| | Small | 8 | 5 | 2048 | 1024 | 14.82M |
| ABIDE & ADHD-200 | Base | 12 | 10 | 2048 | 1024 | 16.80M |
| | Large | 16 | 10 | 3096 | 1024 | 19.19M |
| | X-Large | 20 | 20 | 3096 | 1024 | 21.17M |
| | X-Small | 4 | 4 | 1024 | 512 | 8.83M |
| | Small | 8 | 4 | 2048 | 1024 | 16.53M |
| MDD | Base | 12 | 8 | 2048 | 1024 | 18.78M |
| | Large | 16 | 8 | 3096 | 1024 | 21.50M |
| | X-Large | 20 | 16 | 3096 | 1024 | 23.75M |

## D   COMPARABLE METHOD IMPLEMENTATIONS

**BrainNetCNN**, **DHGNN**, and **BrainGNN** are powerful baselines for brain network analysis. We follow the original architecture and decide the best performance by search the convolution channels in $\{32, 64, 128, 256\}$. The three models are implemented as baseline deep learning models for comparison.

**PopGCN** builts a population graph and predict digagnosis phenotype by node classification in a semi-supervised learning manner. Each node is represented by concatenating the vectorized upper

matrix of the brain networks of a subject. Key features were selected by recursive feature elimination and vectorized into a set for each vertex and then concatenated. The number of selected features is searched in $\{400, 1000, 2000\}$. The adjacency matrix was constructed by the phenotype values (i.e, gender, age, and siteID phenotypes) as well as the similarity between node features.

**vallinaTF** and **BrainNetTF** are implemented based on Transformer networks. Due to the limited sample size of fMRI data, these models are implemented with small sizes. The layer number of Transformer is searched in $\{2, 4, 6, 8\}$ and the number of MHSA heads is decided in $\{2, 4, 8\}$. The hidden feature size is searched in $\{512, 1024, 2048\}$.

**MOCO** and **BYOL** are implemented for comparison with our proposed BTIL in terms of self-supervised representation learning. We carry out experiments by using the same Transforer encoder architecture and data augmentation approaches. In comparision, MOCO learns representations from both positive pairs and negative pairs by contrastive learning, while BYOL only implement positive pairs.

**BrainNPT** and **BrainGSLs**. BrainNPT implements ROI-replacing for building disturbance inputs to help the models better understand the intrinsic patterns of the inputs. As is stated in this study, the only mask and prediction method achieves lower performances than BrainNPT. This might be caused by the limited sample size of fMRI data. In terms of this, other types of masking and prediction self-supervised approaches are not applied for comparison, except BrainGSLs. We follow the originally proposed architecture and change the ROI number to 100/112 on our datasets. For BrainNPT, with the remaining 50% probability, the inputs were replaced by 50% ROIs from another brain network. For BrainGSLs, 10% ROIs were randomly masked.

# E    ADDITIONAL EXPERIMENTS AND RESULTS

## E.1    FEATURE AND PREDICTION SIMILARITY ON ADHD-200 AND REST-MDD DATASET

We also evaluated the time-invariant augmentation methods on the ADHD-200, and REST-MDD datasets. The results coincides with those on the ABIDE dataset. With a proper settings, the resting-state brain functional networks achieve similar features and patterns for diagnosis.

## E.2    EVALUATION ON THE TIME-INVARIANT DATA AUGMENTATION FOR BTIL

Based on the data augmentation testing, we performed evaluations on training BTIL. The results are shown in Figure 6. By random dropping a fixed number of timepoints in BOLD signals, the number of 40 timesteps is preferable for all three datasets. And 10 or 20 timestep dropping might contribute to weak augmentation. This tendency also exists in those by dropping some proportion of Timesteps. Overall, 20-40 timesteps or 10%-20% dropping is the optimal.

## E.3    DIAGNOSIS PERFORMANCE

Table 6: Standard deviation on the 10 runs.

| Methods | | ABIDE-I | | | ADHD-200 | | | REST-MDD | | |
|---|---|---|---|---|---|---|---|---|---|---|
| Model | Type | Acc | Spe | Sen | Acc | Spe | Sen | Acc | Spe | Sen |
| BrainNetCNN | CNN | 2.04 | 7.02 | 4.41 | 1.81 | 8.39 | 1.46 | 1.58 | 3.43 | 1.93 |
| DHGNN | GNN | 1.52 | 5.03 | 2.62 | 2.04 | 2.65 | 2.51 | 1.39 | 1.88 | 1.37 |
| BrainGNN | GNN | 2.24 | 3.59 | 2.57 | 2.59 | 4.05 | 2.85 | 1.84 | 8.02 | 1.37 |
| PopGCN | GNN | 1.40 | 3.12 | 1.74 | 1.36 | 2.17 | 2.98 | 1.90 | 1.90 | 2.79 |
| vanillaTF | TF | 1.13 | 5.19 | 4.03 | 1.14 | 5.20 | 1.74 | 1.03 | 3.31 | 1.94 |
| BrainNetTF | TF | 1.16 | 5.62 | 4.38 | 1.14 | 5.20 | 1.74 | 2.38 | 2.42 | 2.92 |
| MOCO-V3 | SSL | 2.25 | 4.98 | 5.51 | 1.29 | 5.25 | 2.25 | 1.55 | 1.12 | 2.76 |
| BYOL | SSL | 2.11 | 4.90 | 2.36 | 0.99 | 8.37 | 2.17 | 1.16 | 0.82 | 2.15 |
| BrainNPT | SSL | 0.98 | 4.62 | 2.89 | 3.50 | 9.32 | 3.73 | 1.31 | 0.97 | 2.03 |
| BrainGSLs | SSL | - | - | - | 2.96 | 5.28 | 5.42 | 2.67 | 3.10 | 1.23 |
| BTIL | SSL | 3.85 | 5.08 | 6.61 | 2.22 | 5.67 | 2.49 | 1.31 | 1.22 | 1.93 |

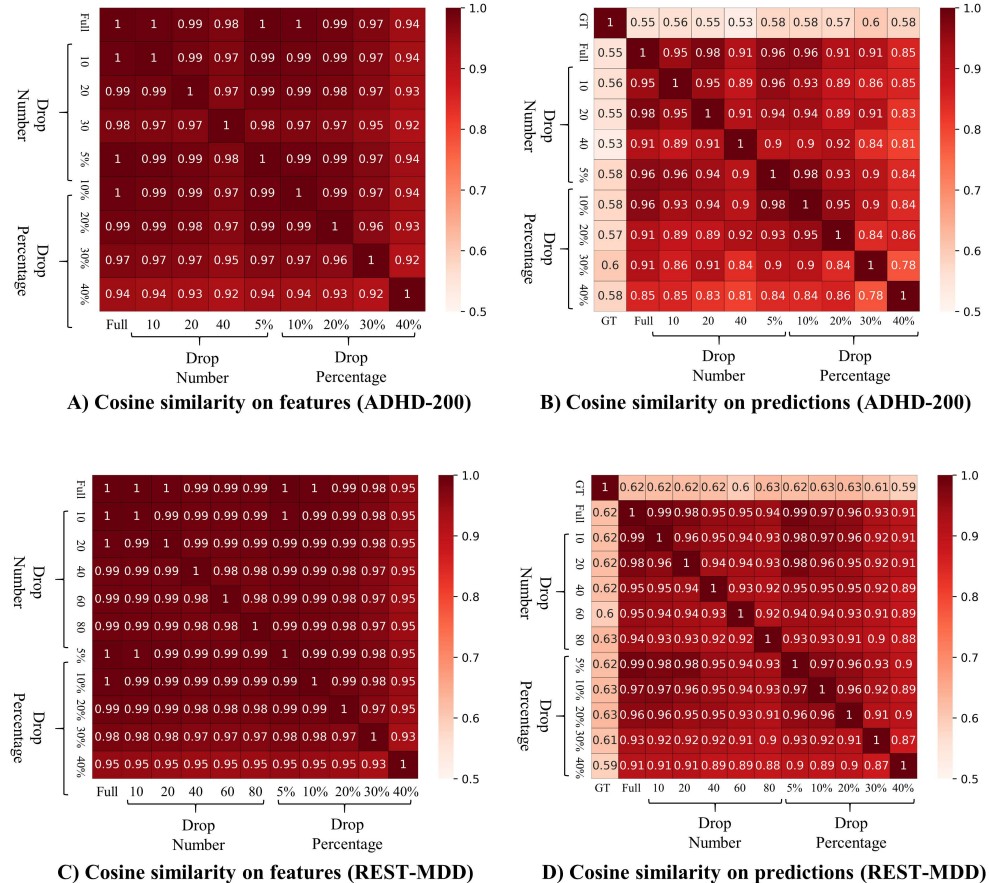

Figure 5: Evaluation on the feature similarity and prediction similarity of time-invariant augmentation on the ADHD-200 and REST-MDD datasets.

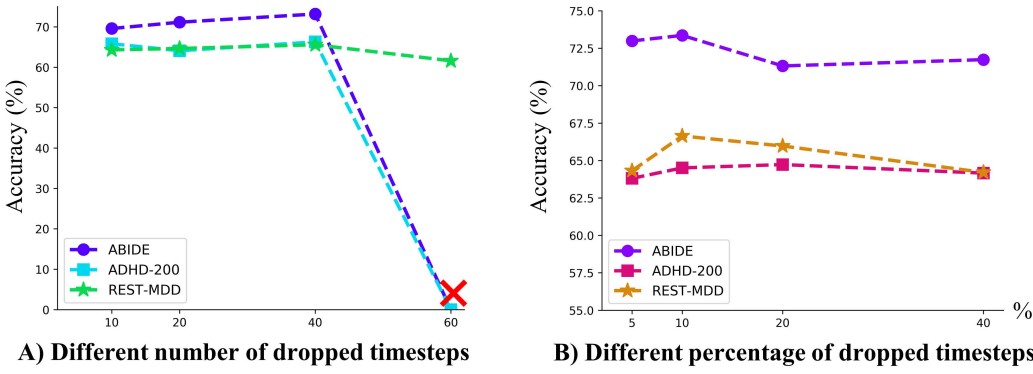

Figure 6: Accuracy performance on different timestep dropping settings.

## E.4 LARGE-SCALE PRETRAINING PERFORMANCE

For pretraining, we acquired an additional dataset specifically for this purpose. The ABIDE-II dataset comprises 677 healthy controls and 559 individuals with ASD. In Figure 7, we present the accuracy performance, with the X-axis denoting the increasing number of training samples. The purple line represents downstream classification accuracy on ABIDE-I, while the pink line represents accuracy on ADHD-200. From the results, we notice that as the number of training samples

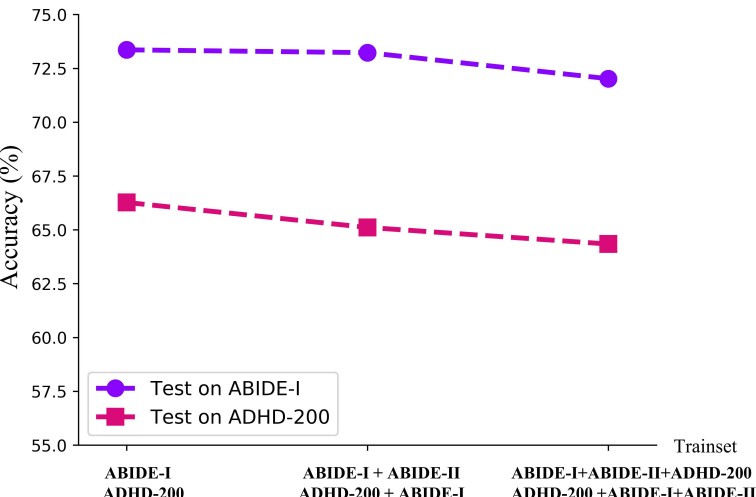

Figure 7: Evaluation on training with broader data. The X-axis demonstrate the training set, while the set set remain fixed for the ABIDE and ADHD-200 datasets.

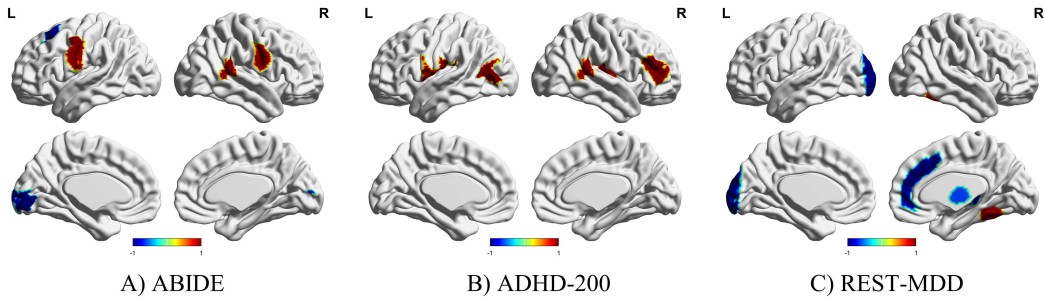

Figure 8: Evaluation on the feature similarity and prediction similarity of time-invariant augmentation.

increases, the performance tends to plateau or exhibit slight fluctuations. Notably, when we include both the ABIDE and ADHD-200 datasets for pretraining, the classification accuracy on ADHD-200 decreases. There are two primary reasons for this. First, the protocols used in these databases differ significantly, resulting in a notable domain gap between the brain networks. Second, the available data samples remain limited, and incorporating more out-of-distribution samples might introduce unwanted noise into the downstream classification process.

## F    BIOLOGICAL EXPLANATION

Our investigation also delved into the crucial brain regions highlighted by the Multi-Head Self-Attention (MHSA). We extracted the attention weights and conducted comparisons between patients and healthy controls (HC) by calculating the difference in weight values, and aggregating them into vectors by summation. In Figure 8, we present the top 5 key features for each of the three datasets. In the results, blue signifies regions that received greater attention from HC, while red indicates regions of heightened attention among patients.

In the case of distinguishing Autism Spectrum Disorder (ASD) and Attention Deficit Hyperactivity Disorder (ADHD) from HC, our analysis identified the Salience Ventricle Attention Network (in the temporal cortex) and the Default Network (in the prefrontal cortex) as key regions. These regions are strongly associated with deficits in executive control and task-irrelevant mental processes, aligning with findings from previous studies Tang et al. (2020); Keehn et al. (2013). Additionally, the visual network and the somatomotor network also emerged as critical regions in both datasets, as they play

pivotal roles in social communication development Lombardo et al. (2019); Marshall et al. (2020). In the case of the REST-MDD dataset, key regions included the paracingulate gyrus, temporal cortex, occipital pole, brain stem, and thalamus. These regions have been identified as important biomarkers for Major Depressive Disorder (MDD) in previous research Yucel et al. (2009); Sacher et al. (2012); Aston et al. (2005); Smoski et al. (2009); Nugent et al. (2013).

