# OpenReview forum: "Uncovering Time-Invariant Latent Representation for Brain Disorder Diagnosis via Self-Supervised Learning"
_ICLR.cc/2024/Conference — ICLR 2024 Conference Withdrawn Submission_

### Official Review · Reviewer_xRi1 · 2023-10-25

**Soundness:** 2 fair
**Presentation:** 1 poor
**Contribution:** 2 fair
**Rating:** 3
**Confidence:** 5

**Summary:**

This paper introduces a self-supervised representation learning in resting-state fMRI for brain disorder diagnosis. For data augmentation, it uses time points masking at random for pseudo-FC generation and ROIs masking for the tasks of classification and reconstruction. The proposed method was tested over three public datasets and performed better than the comparative methods considered in the experiments.

**Strengths:**

The paper focuses on the challenging representation learning in resting-state fMRI.
The experiments were conducted on three public datasets.

**Weaknesses:**

From a methodological perspective, the proposed method exploits the existing ones in the literature and doesn't seem to have a significant contribution. Please explain the difference from the existing methodologies.

It is unclear why the feature representations obtained by the proposed method are time-invariant. Note that the input to the network is in the form of connectivity, already lost the temporal information.

The authors need to check and compare with the paper: Jung et al., “Inter-Regional High-level Relation Learning of Functional Connectivity via Self-Supervision,” MICCAI, 2021.

**Questions:**

It is curious about the performance of adding another linear layer after the Readout and training in an end-to-end manner as a fine-tuning step.

Why does the dimension M of a mask vector need to be smaller than or equal to V?

---

### Official Review · Reviewer_VTCV · 2023-10-29

**Soundness:** 2 fair
**Presentation:** 1 poor
**Contribution:** 2 fair
**Rating:** 3
**Confidence:** 4

**Summary:**

This paper introduces a SSL framework to learn better latent representations of the brain functional connectivity (measured with fMRI), which leads to better downstream tasks. The SSL framework includes 1) time-invariance representation learning (by randomly dropping time points from the input data) and 2) intra-ROI dependency learning (with masked ROI modeling).

**Strengths:**

This paper introduces several exciting concepts in SSL literature (masked modeling, contrastive learning) to brain functional connectivity topic, which had seemed to lack applications of the state-of-the-art techniques. I want to commend the authors on such effort.

**Weaknesses:**

While this paper introduces several novel concepts, there are major weaknesses in its current form of the study and writing that will need to be addressed thoroughly.
- First of all, the definition of "time-invariance" is not defined rigorously. I do not think the authors define the concept clearly, which confuses further understanding of the study. Is time-invariance defined with respect to the entire brain region or only locally? The authors seem to define time-invariance on the level of systems-level (i.e., functional connectivity should be "similar" if the network extracts time-invariant features), but time-invariance could be defined on different temporal/spatial levels in different context.
- The motivation for "time-invariance" was strange/confusing. In page 2, the authors discuss that dFC should be similar to sFC metric. The transition to discussion on demographic factors was not clear (what do these factors have to do with sFC and dFC measures which only concerns timeseries) and then transition to importance of extracting time-invariance representation was also unclear.
- SSL in this context needs to be better motivated - Is SSL the only way to learn "time-invariance"? Are there other methods in the literature that are not SSL, that also works with time-invariance concept?

Writing
- The writing needs to be much clearer. Many notations are not rigorously defined and leaves the reader having to piece together different parts of the paper. For instance, in Section 3.1 (page 3), X is loosely defined as a brain network, to be later defined as the correlation of timeseries matrix S in Section 3.2 (page 4). In page 5, "Readout function", the number 100 appears out of nowhere.
- The description for Masked ROI Modeling is not clear. Maybe it's better to separate Figure 2 into subfigures or different figures to illustrate just the MRM part alone?
- Please be kind to the readers in tiny details. For instance, Table 1 mentions nothing about bold/underlined metrics. In Table 2 and 3, which metrics are they among Acc. / Spe. / Sen. (It's accuracy, but the readers shouldn't be guessing and working hard to figure it out)

**Questions:**

- The pipeline is defined with respect to post-correlation computation X. I am wondering whether it makes more sense to perform the analysis with respect to the raw timeseries S - Pearson correlation is used for X, which I believe might be too restrictive to fully explore the idea on SSL for brain connectivity.
- What would happen if chunk of the timeseries is dropped, instead of randomly dropping the timepoints? Randomly dropping timepoints would only perturb high-frequency information, whereas if chunk was dropped, it would affect low-frequency information of the connectivity, which seems important for resting state fMRI.
- All of the tasks are binary classification tasks - I think it is important to report AUC numbers for all of the tasks.
- I was really confused with Section 4.2 - It is really confusing why the authors wanted to correlate between drop number and drop percentage.
- In Section 4.3, I am not sure if the "time-invariant" representation is indeed crucial as the authors had argued for improved performance compared to others. To really support that claim, the L_latent numbers in Table 2, which presumably uses SSL with just learning the time-invariance representation, should outperform all other baselines in Table 1. Looking at it closely, it seems the outperformance is more down to addition of L_c and L_r, which are for MRM. Therefore, I am not sure if the authors' claims are correct.

---

### Official Review · Reviewer_ivDQ · 2023-10-31

**Soundness:** 3 good
**Presentation:** 2 fair
**Contribution:** 2 fair
**Rating:** 5
**Confidence:** 2

**Summary:**

The paper introduces a new method for representation learning of BOLD signals in a self-supervised manner. They generate positive pairs by masking time points of the temporal signal and obtain views from correlations of those augmented pairs. They apply a masking on the ROI to learn the final representations under a teacher-student scheme. They evaluate the learned embeddings on 3 different datasets (ABIDE, ADHD, Rest-MDD), on which they obtain better performance compared with the literature.

**Strengths:**

- The paper introduces the overall problem and its objectives clearly
- Results are backed by several experiments and compared with a large set of methods from the literature.
- Ablation studies are performed on the different blocks of their architecture, in particular, the role of each loss towards the end goal

**Weaknesses:**

- I had trouble understanding the problem statement initially: X is already a correlation matrix here, while we would expect the problem to predict $ y $ from the BOLD signal. The initial steps from Figure 1 (including the timepoint masking) are occulted here while being an important part of the training process.
- The claim "The regional distinguishing capability is particularly valuable when dealing with disorders
exhibiting significant intra-class variations" (Introduction) feels unsupported in the rest of the paper. Have you measured the effect of $L_c$ on specific disease identification (compared to others with lower intra-class variations)?
- The `metrics` subparagraph indicates that multiple runs were performed, but I couldn't find variance among runs reported.

- Missing citations:
  - Time point and region masking on resting-state BOLD signal:  Thomas, A., Ré, C., & Poldrack, R. (2022). Self-supervised learning of brain dynamics from broad neuroimaging data. Advances in Neural Information Processing Systems, 35, 21255-21269.

**Questions:**

- Could you elaborate on why the latent embeddings obtained would be "disease-specific". It feels that those latent embeddings has no incentive or knowledge towards pushing together disease subjects?
- The readout function looks obscure to me. Could you highlight some references (and include them in the article) or better describe how the feature reduction is performed?
- Why would the readout function be subject-specific? It seems that the readout is optimized against $ O $ from each view, without any knowledge of different views belonging to a same subject?
- I would be interested to know if you investigated learning representations from healthy patients to apply on disease classification. Is there a significant gain by learning the representation model on subjects with said diseases?

---

### Official Review · Reviewer_8QTK · 2023-10-31

**Soundness:** 3 good
**Presentation:** 2 fair
**Contribution:** 1 poor
**Rating:** 3
**Confidence:** 4

**Summary:**

The manuscript presents a framework that combines a data-augmentation technique for fMRI data with self-supervised learning to generate latent representations of functional connectivity matrices. These latent representations were shown to have better discriminative power in three neuroimaging studies to differentiate healthy controls and diseased cohorts.

**Strengths:**

1. The framework was evaluated on 3 different datasets; all 3 are considered relatively large datasets in neuroimaging research.

2. There are multiple ablation studies investigating contributions from different loss terms and hyperparameters.

**Weaknesses:**

1. The proposal method seems incremental; Masking time points is a standard way to augment fMRI data; e.g. in "Identifying autism from resting-state fMRI using long short-term memory networks". Other related techniques, such as cropping the time series and sliding-window-based augmentation, are in the same vein.

2. The "Masked ROI distinguishing" section is unclear. The variables ($x_i$, $o_i$, $r_i$) are defined loosely and the figure does not help understanding.

3. The incremental accuracy of the classification without proof of statistical significance will unlikely bring additional clinical impact.

**Questions:**

1. What if we simply used the augmented data to directly train the classifier? With this masking strategy, we can generate almost infinite number of connectivity matrices.